# A novel use of HIV surveillance and court data to understand and improve care among a population of people with HIV experiencing criminal charges in North Carolina 2017–2020

Elizabeth C. Arant[1]*, Andrew L. Kavee[2], Brad Wheeler[3], Bonnie E. Shook-Sa[4], Erika Samoff[3], David L. Rosen[1]

**1** Division of Infectious Diseases, Department of Medicine, School of Medicine, University of North Carolina at Chapel Hill, North Carolina, **2** Sheps Center for Health Services Research at University of North Carolina at Chapel Hill, North Carolina, **3** North Carolina Department of Health and Human Services, Raleigh, North Carolina, **4** Department of Biostatistics, Gillings School of Global Public Health, University of North Carolina at Chapel Hill, Chapel Hill, North Carolina

* Elizabeth.arant@unchealth.unc.edu

**Prior meetings:** This work has been presented in poster form at IDWeek 2023 in Washington, DC

## Abstract

### Objectives

To enumerate and describe the population of people with HIV (PWH) with criminal charges and to estimate associations between charges and HIV outcomes. We hypothesized that being charged in the court system will be associated with declines in viral suppression.

### Methods

We linked statewide North Carolina (NC) criminal court records to confidential HIV records (both 2017-2020) to identify a population of defendants with diagnosed HIV. We used generalized estimating equations to examine changes in viral suppression (outcome) in the time 12-month periods pre- and post-criminal charges (exposure), adjusting for other demographic and legal system factors.

### Results

9,534 PWH experienced criminal charges. Compared to others with charges, PWH were more likely to be male and report Black race. The median duration of unresolved charges was longer for PWH. A slightly larger proportion of PWH experienced viral suppression in the 12-month post-charge period compared to the pre-charge period (72% vs 70%, p < 0.05). Similarly, when adjusting for demographic factors, the 12-month period following resolution of charges was modestly associated with an increased likelihood of viral suppression (aRR 1.03, 95% confidence interval [CI] 1.02-1.04) compared to the pre-charge period.

**Data availability statement:** The data are not publicly available due their containing information that could compromise the privacy of research participants. Additionally, elements of the linked database are owned by a third-party organization, the North Carolina Department of Public Health (NCDPH). Per our Data Use Agreement with NCDPH, the data cannot be shared. Data requests can be submitted to the UNC IRB at irb_questions@unc.edu for researchers who meet the criteria for access to confidential data.

**Funding:** This work was supported by the National Institute of Allergy and Infectious Diseases (T32AI007001 to EA; R01A129731 DR; and P30AI050410 to EA and DR). The funders had no role in study design, data collection and analysis, decision to publish, or preparation of the manuscript.

**Competing interests:** The authors have declared that no competing interests exist.

## Conclusions

A significant portion of PWH in NC had criminal charges during a three-year period, and these charges went unresolved for a longer time than those without HIV. There was a modest statewide increase in viral suppression in the 12-month period after resolution of charges. Considering the exploratory nature of study, the modest association between charges and viral suppression should not be interpreted causally. In contrast to our expectation, we did not find any evidence that charges were associated with a *decline* in viral suppression. We suggest future lines of research to improve upon this exploratory analysis and area of study.

## Introduction

Criminal system involvement in the United States (US) is common, with about 2.2 million people incarcerated in state and federal prisons and jails in a given day in 2019 [1]. Formerly incarcerated people often have difficulty accessing and utilizing healthcare, resulting in negative impacts on physical and mental health [2,3]. For people with HIV (PWH) specifically, the period following prison or jail incarceration has been associated with loss of viral suppression and decreased engagement in HIV care [4,5].

Although millions of people are incarcerated in the US annually, this population represents a fraction of all Americans who have some type of criminal record [6]. Compared to incarcerations, less is known about the effect of lower-level legal involvement. Others have hypothesized about the burden that arrests, charges, conviction, and sentencing have on physical and emotional health [7–9]. However, little is known about the impact of criminal charges in the absence of incarceration on HIV outcomes.

The goal of the current study was to enumerate and describe the population of PWH who face charges within the criminal legal system and to estimate the association between this involvement and HIV outcomes. To do so, we used a de-identified linked database primarily composed of longitudinal North Carolina (NC) court records and confidential state HIV diagnoses and viral load records. Our central hypothesis is that being charged in the court system will be associated with declines in viral suppression, as court involvement could have a direct negative impact on disease management such as accessing care and adhering to medications; court involvement may also serve as an indicator for social instability and other behaviors that can impede disease management. This analysis represents an important step in describing details about criminal legal involvement for PWH in NC and to better understand the relationship between this involvement and health for PWH.

## Methods

### Data source and charge definition

We utilized data from linked NC court, jail, and HIV surveillance records. This data linkage process has been described previously [10]. In summary, to identify a cohort of adult (aged ≥ 18) PWH in NC charged in criminal court between January 2017- February 2020, we linked the following datasets: statewide court records, jail incarceration records from 26 of 100 counties, and statewide confidential department of public health (DPH) records of all NC PWH. We obtained court records from the NC Administration Office of the Court. We derived jail incarceration histories using a technique called web-scraping, in which automated 'bots' collect incarceration data from public jail rosters. We then linked jail and court records

using identifiers and provided these records to the NC DPH. NC DPH personnel linked these records to the state's confidential HIV records, (using first and last name, DOB, and when available, last 4-digits of social security number), which included statewide viral load (VL) laboratory results and dates. NC DPH then provided us with linked and deidentified court and incarceration records, from which we estimated the number of adult PWH with criminal court records in all 100 NC counties. We used this de-identified database of defendants, both with and without HIV, as the data source in our analysis. The court records assign a unique identifier based on the defendant's name and the county where a charge is initiated, therefore it is possible that one unique participant could be assigned multiple court identifiers if he/she has cases in multiple counties or is identified under a different name. For defendants with HIV, we addressed this by removing cases where a single DPH record was linked to multiple court records (n = 34). For defendants without HIV, we did not have a linked DPH record and were therefore unable to eliminate redundant identifiers. Since this was a rare occurrence among PWH (i.e., < 0.5% of court records), we assumed that this was similarly rare among those without HIV and therefore unlikely to affect our findings.

In the NC court system, a defendant may face one or more criminal charges arising from a single incident. Individual charges can be adjudicated at the same time or sequentially. People may also incur new charges before existing charges are adjudicated. For the purpose of the current analysis, we considered a charge period to be a continuous period in which a person had at least one unresolved charge.

## Covariates

Both court and HIV record demographic information included DOB, race and ethnicity, and sex. HIV records included self-reported HIV transmission group and HIV VL results and dates. Of note, race and ethnicity data were reported slightly differently in the court and DPH records. Most notably, the court records included Hispanic as a race category (alongside White, Black, Other, and Unknown), whereas DPH included separate variables for race and ethnicity. Among PWH, we quantified the discordance between these two classification schemas (S1 Table). For consistency in our main analysis we reported race-ethnicity characteristics using data from the court records only.

## Outcomes

For the primary outcome of VL suppression, we adopted the NC DPH's definition of VL suppression (<200 copies HIV RNA/ml) and un-suppression (≥200 copies HIV RNA/mL or no record of a VL test within 12 months).

## Statistical analysis

The goal of our study was to identify and describe the population of adult diagnosed PWH in NC charged in criminal court 2017-2020 and to estimate the association between this involvement and HIV viral suppression. We used descriptive statistics to report baseline characteristics, number of discrete periods of non-overlapping charges, and duration of unresolved charge period (days). We compared these characteristics by known HIV status at the time of the first charge during the study period. For PWH who experienced a single discrete period of criminal charges during the period studied (94.1%), we calculated the proportion with viral suppression in the 12 months before initiation of and after resolution (pre-post) of criminal charges, using the VLs closest to the charge period. We included only PWH who were diagnosed at least 12 months prior to charges and had at least 12 months of follow-up following the disposition of charges. We coded PWH without a documented VL as unsuppressed. We

implemented a log-binomial model using generalized estimating equations (GEE) to compare the proportion of viral suppression pre-post charges, adjusting for demographic (age, race, sex) and HIV-specific (transmission category) factors. We reported crude risk ratios (RRs), adjusted risk ratios (aRRs), and 95% confidence intervals (CI's). We stratified the analysis of the entire population by the number of days that charges remained unresolved to determine whether our results differed by duration of unresolved charges.

Although our primary aim was to understand the relationship between having criminal charges and viral suppression in PWH in NC, we recognized that some with charges would have spent time incarcerated. Therefore, among people charged in the 26 counties for which we had jail incarceration data, we conducted a sensitivity analysis examining the association between charges and viral load suppression, comparing stratified estimates of the RR and aRR between those with and without an incarceration record.

## Ethics approval/Consent

This study was approved by the University of North Carolina at Chapel Hill institutional review board #17-0946. The IRB waived the requirements for informed consent. Data were fully anonymized before being accessed by the study team, and authors did not have access to information that could identify individual participants either during or after data collection. Data were accessed for research purposes between 5/1/2/21 and 7/1/2023.

## Results

### Full analysis

A total of 2,436,732 unique defendants experienced criminal charges in NC 2017-2020; 9,534 (0.4%) of these were PWH. Compared to those without HIV (n = 2,427,198), PWH were more likely to self-identify as male (78.8% vs 69.4%). A larger number of PWH with criminal charges were Black compared to those without HIV (77.1% vs 36.8%). PWH had a median age of 41 years (interquartile range [IQR] 29-51) compared to 36 years (IQR 27-47) for those without HIV. A majority of PWH and those without HIV had a single period of criminal charges during the study period (94.1% vs 92.7%). PWH had a longer period of days with unresolved charges compared to those without HIV (median of 68 days vs 37 days). The most common mode of HIV acquisition reported at the time of diagnosis were men who have sex with men (MSM) (45.2%); 28.4% had no transmission information recorded (Table 1).

Of the 9,534 PWH with criminal charges, 3,072 (32%) were excluded from multivariable analysis, because they did not contribute 12 months of observable person-time before and after the period of criminal charges. An additional 35 (0.4%) were excluded due to missing a value for at least one covariate, leaving 6,427 (67%) available for analysis. PWH who had full follow-up periods were similar to those without based on distribution of demographic and HIV-specific characteristics in each demographic category (S2 Table). Among those with complete follow-up periods, 662 (10.2%) had a missing viral load in both the pre- or post-charge period and were therefore coded as unsuppressed.

In an unadjusted comparison, a slightly larger proportion of PWH experienced viral suppression in the post-charge period compared to the pre-charge period (72% vs 70%, $p < 0.05$), corresponding to a RR of 1.03 (95% CI 1.02-1.05). When adjusting for demographic and HIV-specific factors, the period following resolution of charges was similarly associated with an increased risk of viral suppression, aRR 1.03 (95% CI, 1.02-1.04). Factors negatively associated with viral suppression in the multivariable model included Black race (aRR 0.92 [95% CI 0.90-0.95]) compared to White race and self-reported HIV acquisition via heterosexual contact (aRR 0.92 [95% CI 0.88-0.97]) or IDU (aRR 0.85 [95% CI 0.80-0.90]) compared to MSM. Factors positively

**Table 1. Baseline characteristics of people with and without HIV who were charged in NC criminal court, 2017-2020.**

| Group | Non-HIV | HIV |
|---|---|---|
| | (n = 2,427,198) | (n = 9,534) |
| | n (col%) or median (1st-3rd quartiles) | |
| **Sex** | | |
| Male | 1,685,236 (69.4) | 7,516 (78.8) |
| Female | 741,962 (30.6) | 2,018 (21.2) |
| **Race** | | |
| White | 1,067,557 (43.9) | 1,860 (19.5) |
| Black | 894,930 (36.8) | 7,353 (77.1) |
| Hispanic | 330,205 (13.6) | 133 (1.4) |
| Other or Unknown | 136,594 (5.6) | 168 (1.8) |
| **Age (years)** | 36 (27-47) | 41 (29-51) |
| **Total number of cases** | | |
| 1 | 2,252,747 (92.7) | 8,969 (94.1) |
| 2 | 157,972 (6.5) | 549 (5.8) |
| 3 or greater | 19,567 (0.8) | 16 (0.2) |
| **Charge days** | 37 (1-129) | 68 (1-209) |
| **HIV transmission group** | | |
| MSM | | 4,306 (45.2) |
| Unknown | | 2,712 (28.4) |
| Heterosexual contact | | 1,367 (14.3) |
| IDU | | 722 (7.6) |
| MSM & IDU | | 358 (3.8) |
| Other | | 69 (0.7) |

Abbreviations: NC, North Carolina; MSM, men who have sex with men; IDU, injection drug use; IQR interquartile range.

associated with viral suppression included those aged 60-69 (aRR 1.27 [95% CI 1.19-1.34)] compared to those aged 18-29 (Table 2). When stratifying the analysis by duration of unresolved charges, those with the longest period of unresolved charges (300 + days, n = 604 [10%]) were more likely to achieve viral suppression (RR 1.06 [95% CI, 1.01-1.12) compared to those with the shortest period of unresolved charges (0-99 days, n = 4,237 [66%]), RR 1.03 [95% CI, 1.01-1.04], although the proportion with viral suppression was overall lower in both the pre- and post-charge periods among those with charges 300 + days (64.4% and 68.1%, respectively), S3 Table.

## Sensitivity analyses

To better understand whether jail incarceration affected the relationship between charge period and viral suppression, the above analyses were conducted in a subset of PWH with charges in the 26 NC counties where jail incarceration data were available. These results were stratified by jail incarceration status.

Among these counties, 157,049 people experienced criminal charges during 2017-2020, of which 1,304 (0.8%) were PWH. There were some demographic differences among this limited population of PWH and the overall population of PWH with criminal charges (S4 Table). We found that when stratifying PWH by jail incarceration status, the relationship between charge period and viral load was similar in both groups. This was true both for unadjusted analyses (RR: 0.98 [95% CI 0.89-1.08] among jail incarcerated, RR: 0.97 [95% CI 0.97 (0.99-1.09)]

**Table 2. Association**[1] **between criminal charge period, demographics, and HIV viral suppression, NC 2017-2020 (n = 6,427)**[2]**. Analysis including data from all NC counties.**

| Characteristic | Adjusted RR (95% CI)[3] |
|---|---|
| **Time period** | |
| 12 months pre charge | |
| 12 months post charge | 1.03 (1.02-1.04) |
| **Age (years)** | |
| 18-29 | Reference |
| 30-39 | 1.03 (0.98-1.08) |
| 40-49 | 1.17 (1.11-1.22) |
| 50-59 | 1.24 (1.19-1.3) |
| 60-69 | 1.27 (1.19-1.34) |
| 70 and older | 1.19 (0.99-1.43) |
| **Race/Ethnicity** | |
| White | |
| Black | 0.92 (0.90-0.95) |
| Hispanic | 1.05 (0.96-1.16) |
| Other | 0.95 (0.85-1.06) |
| **Sex** | |
| Male | |
| Female | 1.02 (0.98-1.06) |
| **HIV Transmission Category** | |
| MSM | |
| Heterosexual | 0.92 (0.88-0.97) |
| IDU | 0.85 (0.80-0.90) |
| MSM and IDU | 0.88 (0.82-0.95) |
| Other | 0.87 (0.69-0.91) |

[1]Multivariable log-binomial estimates.

[2]Only individuals who contributed person-time to both pre- and post-charge periods and who had a single period of criminal charges were included in mode.l

[3]Adjusted RRs are adjusted for all the variables in the table. Abbreviations: PWH, People Living with HIV; NC, North Carolina; RR, risk ratio; CI, confidence interval; HIV, human immunodeficiency virus; MSM, men who have sex with men; IDU, injection drug use. Definitions: Viral Suppression: < 200 copies of HIV RNA per milliliter of blood

among those without jail incarceration) and adjusted analyses (S5 Table). We note that for all estimates, the association was not statistically significant and differed in direction from the estimated associations in our main analysis.

## Discussion

A large proportion of people in the US have some type of criminal record, and history of even a minor criminal record can have negative impacts on physical and mental health [6,8,9,11]. Most people facing criminal charges are not incarcerated or are incarcerated for brief periods. Through an innovative linkage of HIV surveillance and court data, we identified a population of diagnosed PWH with any criminal charges and examined the relationship between having criminal charges—including those with no or short incarcerations—with HIV outcomes. We found that over a three-year period nearly 10,000 PWH in NC experienced criminal charges, which in the year 2019, corresponded with about 1 in 5 PWH. The resolution of charges typically required 80% more time among PWH than

among those without HIV. Contrary to our expectation, we did not find a post-charge decline in viral load suppression.

This study represents one of the largest ever of legal involvement among PWH. While one study evaluated linkage to care in almost 10,000 PWH following release from US jails [12], other studies of incarcerated or formerly incarcerated PWH have been smaller [4,5,13]. No other study has evaluated associations between criminal charges without incarceration and HIV outcomes.

Existing studies of HIV among justice-involved populations have shown that the prevalence of HIV in jails and prisons is about three to five times that of the general US population (with most HIV acquisitions occurring pre-incarceration)[14,15]. Compared to studies of incarcerated populations, we found that the prevalence of HIV among those with criminal charges (0.4%) was similar to the prevalence of HIV in the general population [16]. Nevertheless, a relatively large number of PWH experienced charges. PWH experience a disproportional amount of social and economic inequality [17–19], and our finding that charges among PWH took longer to resolve compared to those without HIV could be explained by PWH's more modest resources. Other possibilities are that PWH had more serious charges that required longer to resolve, or that they were disproportionately charged in court districts that were slower to process cases. Regardless of the ultimate explanation, our findings suggest that PWH may, as a population, benefit from healthcare practice models that include a medical-legal partnership (MLP) [20].

MLPs are healthcare delivery innovations that embed civil legal aid expertise into the health care team and focus on prevention by addressing upstream social and legal problems that affect health [21]. Such partnerships could be expanded to include criminal legal aid, particularly in locations and patient populations with frequent or prolonged legal contact. A group in Rhode Island has expanded the MLP model to include criminal cases, and they have reported positive experiences [22]. While special considerations would need to be made for PWH given the sensitivity and stigma that an HIV diagnosis carries in society, this novel criminal MLP model serves as an example of a potential multidisciplinary support mechanism for PWH experiencing criminal charges, and such services could potentially be integrated into existing HIV services.

Racial disparities pervade the criminal justice system; Black Americans are disproportionately arrested and convicted and are given harsher sentences compared to Whites who commit the same crimes [23,24]. There are also significant racial disparities in the HIV epidemic [18,25]. Systemic racism, HIV stigma, and economic and systemic barriers to health care that underlay disparities related to HIV care likely compound the existing racial disparities within the criminal justice system. In our study, we found a higher proportion of Black PWH with criminal charges compared to the general population of PWH in NC [16]. While not unexpected, our finding of a disproportionate number of Black defendants with HIV suggests that this compounding of racial disparities affects even early legal-system involvement for PWH. Combatting and and addressing systemic racism both in the HIV epidemic and in the criminal justice system are crucial to addressing disparities in these spaces, and HIV clinics could benefit from the support that an MLP model may provide to address upstream barriers related to systemic racism.

Our finding of a very modest improvement in the proportion of viral suppression following the period of criminal charges was unexpected. The crude results reflect that this was indeed the experience of the population. Although the multivariable estimates also reflected a modest improvement in viral load during the post-charge period (RR = 1.03), in considering the small magnitude of the effect and the possibility of unmeasured confounding, this findings should be interpreted with caution and warrants more nuanced analyses in the future. Reframing

the findings, we believe that it is more relevant to highlight that—contrary to our expectation—the post-charge period was not associated with a *decline* in viral load suppression. So while charges may have a multitude of deleterious effects on the health of PWH, based on our findings, a worsening viral load was not among them. However, given the large population of PWH impacted by criminal charges and the novelty of the current study, we recommend that future studies in other states be conducted to confirm our findings.

Our findings add to the existing literature addressing HIV outcomes among PWH in the criminal legal system, which primarily focuses on the effect of prison or jail incarcerations on viral suppression. Most studies show that virologic suppression improves during the prison incarceration period, likely due to access to HIV care and mental health services and a structured daily routine [26]. While HIV care and ART are less consistently available during jail incarcerations, jail lengths of stay are typically shorter (median of 3 days [27]), therefore the peri-incarceration period itself likely plays less of a role in viral suppression. Jail incarceration data were only available to us for a subset of counties. In these counties, a higher proportion of PWH with criminal charges were incarcerated compared to those without HIV (75.8% vs 67.8%). In our sensitivity analysis, the association between charge period and viral suppression was similar among those with and without record of a jail incarceration; this consistency suggests that incarceration did not modify the overall association between charge period and viral suppression. The direction of the charge period-viral suppression association in our sensitivity analysis was in the opposite direction of the association in our main analysis, suggesting that this association may vary by county. At the same time, the small effect sizes and their lack of statistical significance from these sensitivity analyses suggest that these findings must be interpreted with great caution.

While relatively minor offenses usually do not incur other post-sentencing sanctions, these low-level forms of justice system contact can negatively impact physical and mental health throughout the continuum of contact, particularly for those already experiencing social and health disparities [8,11]. We anticipated that the period following criminal charges would be associated with a decline in the proportion of PWH with viral suppression; instead, we found a small increase in the proportion of those with viral suppression following the period of criminal charges. It is likely that most of the criminal charges in our study were minor offenses or misdemeanors, such as traffic violations, that may have had little impact on HIV outcomes. When stratifying the analysis by duration of unresolved charges, however, those with the longest period of charges (300 + days) were more likely to achieve viral suppression compared to those with shorter charge periods. More investigation is needed to understand the relationship between type and duration of charges as they relate to HIV outcomes.

The proportion of PWH with viral suppression in our population was in between that of all PWH in NC in 2020 (66%) and those considered in care (85%) [28]. Given that a relatively small proportion (10%) of our population had missing viral load data, it may be more meaningful to compare the rate of viral suppression to the "in-care" population of PWH in NC. The justice-involved PWH in our study had a lower proportion of viral suppression compared to in-care PWH in NC, which is more consistent with our hypotheses regarding the effect of lower-level legal involvement on HIV outcomes. An alternative etiology relates to limitations of HIV surveillance data. It is possible that individuals who are still in care in another state or who have died may continue to contribute to the surveillance denominator, leading to an underestimation of the true rate of viral suppression. The NC DPH mitigates this issue using active surveillance among those originally identified as out-of-care. For example, in 2020, 26% of people originally deemed out-of-care were identified by NC DPH as living in another state or as having died [28]. Nevertheless, unidentified out-migrations and deaths may still impact our comparisons of viral suppression among PWH with and with court involvement.

A limitation of our study is its reliance on surveillance data. Limitations of this approach have been discussed previously [29–31]. In our full analysis, about 10% of the population had missing VL data in both the pre- and post- charge periods. Prior literature suggests that, there is underreporting of VL to state surveillance systems [32]. As a result, viral suppression is underestimated. While there was considerable variation with laboratory reporting compliance in NC when the surveillance database was first created [33], we believe that modern laboratory platforms and automated reporting have minimized missing data during the study period. Clinicians may decide to reduce VL testing frequency due to more effective and less toxic ART regimens, patient convenience, or cost [34]. Nevertheless, with our use of a full 12-months for our pre- and post-charge time frame, we believe that these factors resulted in "missing" VLs infrequently.

As discussed above, some PWH may receive VL monitoring in different states due to convenience or migration, and excluding or assuming a "worst case" HIV-related outcome for these individuals may result in an overestimation of those who are either out of care or not virally suppressed [35]. Assuming that those without VL data were not virally suppressed may underestimate the proportion of PWH with viral suppression. When excluding those with a missing VL from the analysis, our estimates were similar to estimates obtained when applying the missing data assumption, suggesting that coding those with missing VL data as unsuppressed had little impact on our results.

Another important limitation is that our modeling did not control for important covariates associated with viral suppression (e.g., duration of HIV infection, duration of ART exposure, nadir CD4 cell count) or the legal process (e.g., county, type of charge). It is possible that incusion of these variables would have allowed estimation of our main effect with less bias. Models incorporating these variables should be considered in the future.

## Conclusions and public health implications

Legal involvement, from minor offenses to arrests and charges leading to incarceration, is pervasive in the US. While more is known about the negative impact of incarceration on HIV-related outcomes for PWH, a better understanding of the impact of lower-level legal-involvement in this population is needed. In our study, a significant portion of PWH in NC had criminal charges during a three-year period, and these charges went unresolved for a longer time than those without HIV. Although charges did not appear to have a deleterious effect on viral suppression, additional studies should be conducted in other states to examine the generalizability of our findings. An important step would be to develop statistical models with more nuanced clinical and legal information. Nevertheless, our preliminary findings raise questions regarding whether PWH have appropriate access to legal services. Greater access to legal care could be facilitated by developing medical-legal partnerships at HIV care sites. Qualitative investigation of the experiences that PWH with criminal charges have with legal representation and their healthcare teams would strengthen our understanding about interventions that may be most beneficial to improving HIV care for this population. Given that a significant number of PWH across the US likely experience criminal charges, ongoing work in this space is crucial for informing care for this population.

## Supporting information

**S1 Table. Department of public health versus Court record coding of race/ethnicity.** (PDF)

**S2 Table. Baseline characteristics of people with HIV without full follow-up periods (n = 2,847).** (PDF)

**S3 Table. Association between criminal charge period and duration of unresolved charges NC 2017–2020: Results of multivariable log-binomial model (n = 6,427)2 Analysis including data from all NC counties.**
(PDF)

**S4 Table. Baseline characteristics of people with and without HIV who charged in criminal court in NC between 2017–2020. Data limited to 26 counties with jail incarceration data available.**
(PDF)

**S5 Table. Viral suppression outcome among people with HIV (PWH) in NC with a single criminal charge between 2017-2020: results of multivariable log-binomial model limited to NC counties with available jail incarceration data.**
(PDF)

## Author contributions

**Conceptualization:** Elizabeth C. Arant, David L. Rosen.

**Data curation:** Andrew L. Kavee, David L. Rosen.

**Formal analysis:** Elizabeth C. Arant, David L. Rosen.

**Funding acquisition:** Elizabeth C. Arant, David L. Rosen.

**Writing – original draft:** Elizabeth C. Arant.

**Writing – review & editing:** Elizabeth C. Arant, Andrew L. Kavee, Brad Wheeler, Bonnie E. Shook-Sa, Erika Samoff, David L. Rosen.

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
