## [Decision Letter · Decision Letter 0]

2 Aug 2024

PONE-D-24-13276A novel use of HIV surveillance and court data to understand and improve care among a population of people with HIV experiencing criminal charges in North Carolina 2017-2020PLOS ONE

Dear Dr. Arant,

Thank you for submitting your manuscript to PLOS ONE. After careful consideration, we feel that it has merit but does not fully meet PLOS ONE’s publication criteria as it currently stands. Therefore, we invite you to submit a revised version of the manuscript that addresses the points raised during the review process.

We look forward to receiving your revised manuscript.

Kind regards,

Sepiso Kenias Masenga, PhD

Academic Editor

PLOS ONE

“This work was funded by the National Institute of Allergy and Infectious Diseases at the National Institutes of Health ([NIH] Grant Number T32AI007001 to E.C.A and R01A129731 to D.L.R).  This research was supported by the University of North Carolina at Chapel Hill Center For AIDS Research (CFAR), an NIH funded program P30AI050410.”

4. We notice that your supplementary figures are uploaded with the file type 'Figure'. Please amend the file type to 'Supporting Information'. Please ensure that each Supporting Information file has a legend listed in the manuscript after the references list.

Reviewers' comments:

Reviewer's Responses to Questions

**Comments to the Author**

1. Is the manuscript technically sound, and do the data support the conclusions?

Reviewer #1: Yes

Reviewer #2: Yes

2. Has the statistical analysis been performed appropriately and rigorously? 

Reviewer #1: Yes

Reviewer #2: Yes

3. Have the authors made all data underlying the findings in their manuscript fully available?

Reviewer #1: Yes

Reviewer #2: Yes

4. Is the manuscript presented in an intelligible fashion and written in standard English?

Reviewer #1: Yes

Reviewer #2: Yes

5. Review Comments to the Author

Reviewer #1: This is a very interesting paper that uses linked criminal records and HIV surveillance data to examine viral suppression outcomes before and after criminal charges in NC. The linked dataset is very unique, and the analysis addressed an important public health topic. Overall, the paper is clearly written and I did not identify any major flaws. Minor comments for authors to consider:

A result contradicting the original hypothesis has been found: PWH had slightly better viral suppression outcomes after criminal charges than before. This is a very interesting result. The authors included some nice discussion about this. I am wondering is it possible that this association was observed due to any uncontrolled confounder? If so, what confounder do the authors think it could be?

Have the authors considered using the first viral load test after the criminal charges to see if a lower proportion of viral suppression can be found compared to before? A shorter time window may better capture the immediate impact of criminal charges on HIV viral suppression outcomes. People could have recovered from the mental stress related to criminal charges in the next 12 months, especially if most of the charges were for minor offenses.

Reviewer #2: Title: A novel use of HIV surveillance and court data to understand and improve care among a population of people with HIV experiencing criminal charges in North Carolina 2017- 2020

Study summary

The study by Arant et al. investigates the impact of criminal legal system involvement on HIV outcomes among people with HIV (PWH) in North Carolina (NC). Using a linked database of NC court records, jail incarceration records, and state HIV records, the study analyzed data from PWH charged with offenses between January 2017 and February 2020. The study found that while PWH had longer unresolved charge durations, viral suppression rates modestly increased post-charge resolution. Arant et al.’s findings suggest potential benefits of integrating medical-legal partnerships into healthcare delivery for PWH facing detention.

Major strengths of the study

The study was well-designed and conducted, and the report is also well written.

Minor comments

1. In lines 197-199 the authors mention that “When adjusting for demographic and HIV-specific factors, the period following resolution of charges was similarly associated with an increased risk of viral suppression.” Since viral load suppression is a positive outcome, I suggest the authors use "likelihood" or "odds" instead of "risk" which is associated with a negative outcome.

6. PLOS authors have the option to publish the peer review history of their article (what does this mean? ). If published, this will include your full peer review and any attached files.

**Do you want your identity to be public for this peer review?** For information about this choice, including consent withdrawal, please see our Privacy Policy .

Reviewer #1: No

Reviewer #2: **Yes: ** Lweendo Muchaili

---

## [Author Response · Author response to Decision Letter 1]

24 Sep 2024

Dear Reviewers/Editors,

Thank you for your helpful comments and feedback. I have responded to each suggestion below.

Sincerely,

Elizabeth Arant

a. Response: I have made the appropriate edits to the headings, figure naming, and citations

a. Response: I have added the comment about the funding role to the cover letter and have moved the funding statement to the cover letter

“This work was funded by the National Institute of Allergy and Infectious Diseases at the National Institutes of Health ([NIH] Grant Number T32AI007001 to E.C.A and R01A129731 to D.L.R). This research was supported by the University of North Carolina at Chapel Hill Center For AIDS Research (CFAR), an NIH funded program P30AI050410.”

a. Response: I have altered the data availability statement in the online submission form

4. We notice that your supplementary figures are uploaded with the file type 'Figure'. Please amend the file type to 'Supporting Information'. Please ensure that each Supporting Information file has a legend listed in the manuscript after the references list.

a. Response: I have included a section in the paper for supplementary information with appropriate file titles. I have also uploaded each supplementary table separately

a. Response: supporting tables have captions at the end of the manuscript and I have updated in-text citations with the new table names

a. I have reviewed my list of citations. While no citations were retracted, I did make some errors in my in-text citations and have made those corrections.

Reviewers' comments:

Reviewer's Responses to Questions

Comments to the Author

1. Is the manuscript technically sound, and do the data support the conclusions?

Reviewer #1: Yes

Reviewer #2: Yes

2. Has the statistical analysis been performed appropriately and rigorously?

Reviewer #1: Yes

Reviewer #2: Yes

3. Have the authors made all data underlying the findings in their manuscript fully available?

Reviewer #1: Yes

Reviewer #2: Yes

4. Is the manuscript presented in an intelligible fashion and written in standard English?

Reviewer #1: Yes

Reviewer #2: Yes

5. Review Comments to the Author

Reviewer #1: This is a very interesting paper that uses linked criminal records and HIV surveillance data to examine viral suppression outcomes before and after criminal charges in NC. The linked dataset is very unique, and the analysis addressed an important public health topic. Overall, the paper is clearly written and I did not identify any major flaws. Minor comments for authors to consider:

A result contradicting the original hypothesis has been found: PWH had slightly better viral suppression outcomes after criminal charges than before. This is a very interesting result. The authors included some nice discussion about this. I am wondering is it possible that this association was observed due to any uncontrolled confounder? If so, what confounder do the authors think it could be?

Response: Thank you for bringing this up. I think the most likely uncontrolled confounder would be the severity of criminal charges, which we did not capture in this analysis. It is likely that those with relatively minor offenses made up the majority of the population of PWH with criminal charges. My thought is that a minor offense had little, if any, effect on viral suppression. Perhaps brief or limited contact with the criminal justice system would serve as prompt for the individual to re-engage in social services that may also improve engagement in HIV care.

Have the authors considered using the first viral load test after the criminal charges to see if a lower proportion of viral suppression can be found compared to before? A shorter time window may better capture the immediate impact of criminal charges on HIV viral suppression outcomes. People could have recovered from the mental stress related to criminal charges in the next 12 months, especially if most of the charges were for minor offenses.

Response: I agree with this, and this point was also brought up when I presented this data at our ID grand rounds. We did use the viral load closest to the time period of the end of the charge, so I do think we captured the first data point post-charge period. One challenging aspect is that charge periods can sometime last for several months (charges take a while to resolve), so even the first VL after the charge period may not represent the impact of that first contact with the criminal justice system.

Reviewer #2: Title: A novel use of HIV surveillance and court data to understand and improve care among a population of people with HIV experiencing criminal charges in North Carolina 2017- 2020

Study summary

The study by Arant et al. investigates the impact of criminal legal system involvement on HIV outcomes among people with HIV (PWH) in North Carolina (NC). Using a linked database of NC court records, jail incarceration records, and state HIV records, the study analyzed data from PWH charged with offenses between January 2017 and February 2020. The study found that while PWH had longer unresolved charge durations, viral suppression rates modestly increased post-charge resolution. Arant et al.’s findings suggest potential benefits of integrating medical-legal partnerships into healthcare delivery for PWH facing detention.

Major strengths of the study

The study was well-designed and conducted, and the report is also well written.

Minor comments

1. In lines 197-199 the authors mention that “When adjusting for demographic and HIV-specific factors, the period following resolution of charges was similarly associated with an increased risk of viral suppression.” Since viral load suppression is a positive outcome, I suggest the authors use "likelihood" or "odds" instead of "risk" which is associated with a negative outcome.

Response: Thank you for pointing this out. I have adjusted the wording in the text

6. PLOS authors have the option to publish the peer review history of their article (what does this mean?). If published, this will include your full peer review and any attached files.

Do you want your identity to be public for this peer review? For information about this choice, including consent withdrawal, please see our Privacy Policy.

Reviewer #1: No

Reviewer #2: Yes: Lweendo Muchaili

---

## [Decision Letter · Decision Letter 1]

24 Dec 2024

PONE-D-24-13276R1A novel use of HIV surveillance and court data to understand and improve care among a population of people with HIV experiencing criminal charges in North Carolina 2017-2020PLOS ONE

Dear Dr. Arant,

Thank you for submitting your manuscript to PLOS ONE. After careful consideration, we feel that it has merit but does not fully meet PLOS ONE’s publication criteria as it currently stands. Therefore, we invite you to submit a revised version of the manuscript that addresses the points raised during the review process.

We look forward to receiving your revised manuscript.

Kind regards,

Sepiso K. Masenga, PhD

Academic Editor

PLOS ONE

Journal Requirements:

**Additional Editor Comments:**

Please take time to address comments from reviewer 3 carefully.

Reviewers' comments:

Reviewer's Responses to Questions

**Comments to the Author**

1. If the authors have adequately addressed your comments raised in a previous round of review and you feel that this manuscript is now acceptable for publication, you may indicate that here to bypass the “Comments to the Author” section, enter your conflict of interest statement in the “Confidential to Editor” section, and submit your "Accept" recommendation.

Reviewer #2: All comments have been addressed

Reviewer #3: (No Response)

2. Is the manuscript technically sound, and do the data support the conclusions?

Reviewer #2: (No Response)

Reviewer #3: No

3. Has the statistical analysis been performed appropriately and rigorously? 

Reviewer #2: (No Response)

Reviewer #3: No

4. Have the authors made all data underlying the findings in their manuscript fully available?

Reviewer #2: (No Response)

Reviewer #3: Yes

5. Is the manuscript presented in an intelligible fashion and written in standard English?

Reviewer #2: (No Response)

Reviewer #3: Yes

6. Review Comments to the Author

Reviewer #2: (No Response)

Reviewer #3: The study takes an innovative approach by exploring a unique intersection of legal and health systems, shedding light on systemic disparities that disproportionately affect PWH. However, despite the study’s strengths, I feel the findings may be on limited clinical relevance. The authors might want to explore more on gathering data to explore associations with between legal systems and VL suppression or focus on different outcomes which might be relevant. My detailed review is as outlined below.

Abstract

The objective states ‘to enumerate the population of people with HIV (PWH) with criminal charges and to estimate associations between charges and HIV outcomes" but the conclusion seems to diverge from the focus on HIV outcomes. Instead, it highlights unresolved criminal charges and access to legal services, which does not directly address the aim of investigating the relationship between criminal charges and HIV-related health outcomes like viral suppression.

‘Objective’ is repeated twice.

Results

The results: “the period following resolution of charges was modestly associated with an increased likelihood of viral suppression (aRR 1.03 (95% confidence interval 1.02-1.04) compared to the pre-charge period” is not clear. What do authors mean by the period? There is no direction stated here leaving the reader to guess.

The conclusion should ideally make a clearer connection between criminal charges and their impact on HIV outcomes, such as treatment adherence, viral load suppression (as stated by authors), or other health measures. Instead of discussing unresolved charges or legal access, it would be more appropriate for the conclusion to consider how criminal charges may affect treatment outcomes. For example, you might expect the conclusion to address whether individuals with criminal charges are less likely to have controlled viral loads which is directly tied to the study's aims.

Main body

Again, the focus should ideally be on exploring the possible link between the presence of criminal charges and HIV-related health outcomes, or identifying gaps in healthcare access that could be influencing these outcomes. Which is not highlighted in this research.

While negative results are acceptable, it seems this might be ‘random noise’ especially that it is not highlighted whether the HIV status was known by key personnel who can directly influence the criminal charge outcomes.

Sensitivity analyses were performed but omitted important parameters which might influence the VL outcome.

Eg duration of HIV infection, duration of exposure to ART, baseline VL (either at the time of data of collection or at the time ART initiation). These would give a better picture of the HIV-related factors that might influence VL suppression. (line 174 to 177) HIV specific factors are not well reported and the criteria for these factors is not indicated. Adjusting for confounders should take into consideration the critical confounders.

A 3% difference may not represent a meaningful improvement in viral suppression for PWH at the individual or population level even though it is statistically significant. The small effect size might result from residual confounding rather than an association, as mentioned earlier, this may just be random noise as the authors have not shown that they considered critical HIV-related confounders.

The discussion from line 242 to 275.

I noted the following

The discussion compares the prevalence of HIV among those with charges to the general population but fails to connect this comparison meaningfully to the study's findings.

The emphasis on social and economic inequalities and prolonged charge resolution do not directly address the association between unresolved charges and HIV outcomes, leaving the argument unfocused.

Hypotheses about longer charge resolution times for PWH (e.g., more serious charges, slower court districts) are speculative without supporting data.

The introduction of the Medical-Legal Partnership (MLP) model feels disconnected from the main findings. While MLP is an innovative idea, its relevance to criminal charges and HIV outcomes in this study context is not well-established.

7. PLOS authors have the option to publish the peer review history of their article (what does this mean? ). If published, this will include your full peer review and any attached files.

**Do you want your identity to be public for this peer review?** For information about this choice, including consent withdrawal, please see our Privacy Policy .

Reviewer #2: No

Reviewer #3: No

---

## [Author Response · Author response to Decision Letter 2]

29 Jan 2025

Dear Editor and Reviewers,

Thank you for considering our manuscript titled “A novel use of HIV surveillance and court data to understand and improve care among a population of people with HIV experiencing criminal charges in North Carolina 2017-2020.” We greatly appreciate your time and thoughtful comments to strengthen the manuscript. We note that this is the second round of reviews, and on the first round, both reviewers had favorable assessments with only minor suggestions. One of the original reviewers, Reviewer #2 had provided comments previously and judged that we had adequately addressed their suggestions. We believe that Reviewer #3 was reviewing our article for the first time. Reviewer #3 found our approach to be “innovative” and that the study shed “light on systemic disparities” among PWH. However, the reviewer thought the findings may have limited clinical relevance and they suggested gathering more data and/or focusing on different outcomes. We fully agree that with this exploratory study and the modest association detected, our findings are unlikely to shift clinical practice. As the reviewer conceded in the review, even null results can be informative and so we maintain that the findings add to existing knowledge in the field, particularly given the novel nature of the study. Further, we maintain that our current outcome, viral load suppression, is commonly used in the field and is appropriate for the current study. And finally, we find the reviewer’s suggestion to incorporate more data (i.e. additional variables) into the analysis a near universal suggestion for all research of this type. We agree that it would be helpful if we were able to integrate additional variables into the analysis, and indeed we did conduct sensitivity analyses to examine how our findings may be influenced by incarceration and by missing viral load data. We think that our current analysis is transparent, well performed, and ultimately informative. We are also careful in describing the limitations of the current work and we make suggestions for future research directions. In total, we believe that this is a worthy contribution to the literature. So with much respect, we are not in full agreement with the reviewer’s critiques. That said, we did find several of Reviewer #3’s specific comments extremely helpful, and we have addressed and incorporated them into our revised manuscript, as described below. Please see the reviewer’s comments presented in italics, and our response in the corresponding bulleted points.

Comment/Suggestion #1: Abstract The objective states ‘to enumerate the population of people with HIV (PWH) with criminal charges and to estimate associations between charges and HIV outcomes" but the conclusion seems to diverge from the focus on HIV outcomes. Instead, it highlights unresolved criminal charges and access to legal services, which does not directly address the aim of investigating the relationship between criminal charges and HIV-related health outcomes like viral suppression.

• As we’ve clarified in the abstract, the focus was on describing the size and characteristics of the population and examining the association between charges and viral load. Initially we leaned into addressing the descriptive findings more because the association was so close to the null. Now we provide more space for both.

‘Objective’ is repeated twice.

• Thank you. We have corrected this.

The results: “the period following resolution of charges was modestly associated with an increased likelihood of viral suppression (aRR 1.03 (95% confidence interval 1.02-1.04) compared to the pre-charge period” is not clear. What do authors mean by the period? There is no direction stated here leaving the reader to guess.

• We have clarified that the comparison is between the pre-charge and post-charge time periods. The direction is clearly stated as we found an increase in viral suppression in the post-period compared to the pre-period.

The conclusion should ideally make a clearer connection between criminal charges and their impact on HIV outcomes, such as treatment adherence, viral load suppression (as stated by authors), or other health measures. Instead of discussing unresolved charges or legal access, it would be more appropriate for the conclusion to consider how criminal charges may affect treatment outcomes. For example, you might expect the conclusion to address whether individuals with criminal charges are less likely to have controlled viral loads which is directly tied to the study's aims.

• We agree with your suggestion and have strengthened the connection between the abstract objective and viral load results in the conclusion.

Comment/Suggestion #2: Main body: Again, the focus should ideally be on exploring the possible link between the presence of criminal charges and HIV-related health outcomes, or identifying gaps in healthcare access that could be influencing these outcomes. Which is not highlighted in this research.

• In this exploratory study, we both aimed to describe the population and to address the relationship between charges and viral load. We are now more explicit in the background about the possible role between court involvement and elements of disease management including clinic attendance and adherence. In the discussion, we also address these issues as they relate to findings in carceral settings.

While negative results are acceptable, it seems this might be ‘random noise’ especially that it is not highlighted whether the HIV status was known by key personnel who can directly influence the criminal charge outcomes. Sensitivity analyses were performed but omitted important parameters which might influence the VL outcome. Eg duration of HIV infection, duration of exposure to ART, baseline VL (either at the time of data of collection or at the time ART initiation). These would give a better picture of the HIV-related factors that might influence VL suppression. (line 174 to 177) HIV specific factors are not well reported and the criteria for these factors is not indicated. Adjusting for confounders should take into consideration the critical confounders.

• As part of the limitations section we now address the absence of these variables in our models and the possibility for uncontrolled/unmeasured confounding.

A 3% difference may not represent a meaningful improvement in viral suppression for PWH at the individual or population level even though it is statistically significant. The small effect size might result from residual confounding rather than an association, as mentioned earlier, this may just be random noise as the authors have not shown that they considered critical HIV-related confounders.

• We agree that a 3% pre-post difference may not represent a clinically meaningful improvement in suppression and that our multivariable findings may be subject to uncontrolled confounding. We now address the limitations of the modeling results and make the greater point that contrary to our expectations, the results are nevertheless informative in that we did not find any evidence suggesting that charges were associated with a decline in viral load.

Comment/Suggestion #3: The discussion from line 242 to 275.

I noted the following

The discussion compares the prevalence of HIV among those with charges to the general population but fails to connect this comparison meaningfully to the study's findings.

• We are now explicit in stating that an objective of the study was to describe the population of PWH with criminal charges. With this stated objective, we believe that there is now a clear connection in describing the characteristics of PWH with criminal charges in the context of PWH in the general population. As an aside, we note that we have also expanded our discussion of the viral suppression findings, incorporating many of the reviewer’s excellent points.

The emphasis on social and economic inequalities and prolonged charge resolution do not directly address the association between unresolved charges and HIV outcomes, leaving the argument unfocused.

• One of the most worrisome findings was the longer period of charges faced by PWH. In this context, we believe that a discussion of social and economic inequalities and legal representation for PWH is warranted.

Hypotheses about longer charge resolution times for PWH (e.g., more serious charges, slower court districts) are speculative without supporting data.

• Given that we did not have granular detail about the types of criminal charges available during the analysis, we generated hypotheses to guide next steps. We believe that our informed speculation is appropriate and will spark policy discussion and future inquiry.

The introduction of the Medical-Legal Partnership (MLP) model feels disconnected from the main findings. While MLP is an innovative idea, its relevance to criminal charges and HIV outcomes in this study context is not well-established.

• As addressed above, given the longer charge times for PWH, we believe that addressing the topic of legal representation is appropriate. We note that while MLPs have traditionally been used to address civil legal issues, we discuss a model in which it also addresses criminal charges. When presenting these findings to various groups, we have found that the idea of greater legal representation for PWH has generated much interest as a possible enhancement to the social services traditionally made available to PWH.

• Once again, we truly appreciate your feedback and the opportunity to improve this paper. We are confident that these revisions have strengthened the manuscript, and we look forward to hearing your thoughts on the updated version. Thank you for your time and consideration.

---

## [Editor Report · Decision Letter 2]

9 Feb 2025

A novel use of HIV surveillance and court data to understand and improve care among a population of people with HIV experiencing criminal charges in North Carolina 2017-2020

PONE-D-24-13276R2

Dear Dr. Arant,

We’re pleased to inform you that your manuscript has been judged scientifically suitable for publication and will be formally accepted for publication once it meets all outstanding technical requirements.

Kind regards,

Sepiso K. Masenga, PhD

Academic Editor

PLOS ONE

Additional Editor Comments (optional): I appreciate the effort you took to address the reviewer comments. I have carefully read and considered all responses and comments to arrive at this decision. 
---

## [Editor Report · Acceptance letter]

PONE-D-24-13276R2

PLOS ONE

Dear Dr. Arant,

I'm pleased to inform you that your manuscript has been deemed suitable for publication in PLOS ONE. Congratulations! Your manuscript is now being handed over to our production team.

Kind regards,

on behalf of

Prof. Sepiso K. Masenga

Academic Editor

PLOS ONE